# CD10 Expression Correlates with Earlier Tumour Stages and Left-Sided Tumour Location in Colorectal Cancer but Has No Prognostic Impact in a European Cohort

**DOI:** 10.3390/cancers16081473

**Published:** 2024-04-11

**Authors:** Julia-Kristin Grass, Katharina Grupp, Martina Kluth, Claudia Hube-Magg, Ronald Simon, Marius Kemper, Jakob R. Izbicki, Guido Sauter, Nathaniel Melling

**Affiliations:** 1Department of General, Visceral and Thoracic Surgery, University Medical Center Hamburg-Eppendorf, Martinistraße 52, 20246 Hamburg, Germany; m.kemper@uke.de (M.K.); izbicki@uke.de (J.R.I.); n.melling@uke.de (N.M.); 2Department of Plastic, Reconstructive and Aesthetic Surgery, University Medical Center Hamburg-Eppendorf, Martinistraße 52, 20246 Hamburg, Germany; k.grupp@uke.de; 3Department of Pathology with Sections Molecular Pathology and Cytopathology, University Medical Center Hamburg-Eppendorf, Martinistraße 52, 20246 Hamburg, Germany; m.kluth@uke.de (M.K.); c.hube@uke.de (C.H.-M.); r.simon@uke.de (R.S.); g.sauter@uke.de (G.S.)

**Keywords:** CD10, tissue microarray, immunohistochemistry, colorectal cancer, overall survival, metalloprotease

## Abstract

**Simple Summary:**

Colorectal cancer remains one of the leading causes of cancer-related mortality worldwide and there is an urgent need for prognostic markers to identify aggressive tumours for intensified treatment. CD10 has been controversially discussed as prognosticator for these tumours. However, the published evidence mainly originates from smaller cohorts from one sub-continent. Therefore, this study aims to clarify the role of CD10 in a large European cohort of 1469 colorectal cancer patients. Our data demonstrate that CD10 expression is associated with earlier tumour stages and left-sided tumours, but we can exclude CD10 as a relevant independent prognosticator for colorectal cancer.

**Abstract:**

The role of CD10 expression in colorectal cancer has been controversially discussed in the literature. Some data suggest a predictive capacity for lymph node and liver metastases, thus influencing overall survival (OS) and disease-free survival (DFS). This study aims to analyse the relationship between CD10 expression and overall survival (OS) in a European cohort. To determine the association of CD10 expression with tumour phenotype, molecular features, and prognosis, a tissue microarray of 1469 colorectal carcinomas was analysed using immunohistochemistry and was compared with matched clinicopathologic data. CD10 expression correlated with earlier tumour stages (*p* = 0.017) and left-sided colon cancer (*p* < 0.001). However, no correlation was found between CD10 expression and lymph node involvement (*p* = 0.711), tumour grading (*p* = 0.397), or overall survival (*p* = 0.562). Even in the subgroup analysis of tumour or nodal stage, CD10 did not affect overall survival, although it was significantly associated with p53 and nuclear β-catenin expression (*p* = 0.013 and *p* < 0.001, respectively). CD10 expression correlates with earlier tumour stages, colon cancer location, and indicators of aggressive CRC subtypes. However, we can exclude CD10 as a relevant independent prognosticator for CRC.

## 1. Introduction

Although recent advances have improved outcomes, colorectal cancer (CRC) remains the second leading cause of cancer-related death worldwide [1]. As genetic alterations occur early in tumour progression, molecular profiling of specific tumour markers in the primary tumour could predict the metastatic potential of the tumour. This implies that patients with potentially aggressive tumours may also benefit from adjuvant therapy, even though the tumour has not yet metastasized at the time of surgery.

CD10, also known as the common acute lymphoblastic leukaemia antigen (CALLA), is a transmembrane metallopeptidase that inactivates a variety of biologically active peptides and is encoded by the *membrane metalloendopeptidase (MME)* gene. It is normally expressed in lymphoid progenitor cells, mature neutrophils, and in healthy tissues of the kidney, small intestine, liver, and endometrium [2,3], but not in healthy colorectal tissue [4]. CD10 was originally used to classify subtypes of leukaemia [5,6] and has also been described in lymphoma and stromal cells of invasive mamma carcinomas [7]. It has also been found in lung, colorectal, and prostate cancers, as well as malignant melanoma, and is known to be a predictor of aggressive cancer through extracellular enzymatic degradation and signalling alteration [8,9,10,11]. 

Matrix metallopeptidases (MMPs) such as CD10 play a pivotal role in cancer progression by regulating cancer cell invasion and promoting carcinogenesis. Tumour cell invasion, growth, and metastatic potential are influenced by tumour–stromal interactions. Stromal fibroblasts produce and secrete MMPs, thus providing a suitable environment for invasion and metastatic growth [12,13].

The role of CD10 in colorectal cancer is controversially discussed in the literature, which is currently primarily based on studies conducted in Japan [8,9,14,15,16,17,18,19,20,21,22]. Furthermore, the impact of CD10 expression on patient prognosis and survival remains unresolved, due to conflicting published data [9,17,22,23]. Some studies have proposed a functional role for CD10 in the early stages of CRC carcinogenesis and its adenoma–carcinoma sequences, inducing potentially more aggressive tumour biology with faster progression in advanced stages and distant metastasis [21,22,24].

Molecular markers that predict aggressive behaviour and risk of metastasis are crucial for further optimizing CRC management. Finally, to clarify the role of CD10 expression in CRC, we analysed a large European cohort using a tissue microarray of 1469 colorectal carcinomas and corresponding histopathological and clinical data.

## 2. Materials and Methods

### 2.1. Patients

Consecutive patients undergoing radical surgery for CRC at the University Hospital of Basel, Switzerland, and the University Medical Center, Hamburg-Eppendorf, Germany, were included. Histopathological and clinical data were collected and analysed. Follow-up data were obtained from local cancer register boards or attending physicians. For statistical analyses, the tumour localizations were grouped as follows: right-sided cancer (cecum and ascending colon), cancer of the transverse colon, cancer of the left-sided colon (descending colon, sigmoid colon), and rectum. The usage of tissue microarray (TMA) technology allowed for efficient analysis at the protein level. All samples were analysed according to a standard procedure, including complete embedding of the entire specimen for histological analysis. The study was approved by the Hamburg Ethics Committee and performed in accordance with the Declaration of Helsinki. The use of routinely archived, formalin-fixed residual tissue samples from patients for research is permitted by law and does not require written informed consent (HmbKHG, §12.1).

### 2.2. Tissue Microarray Manufacturing

TMAs were constructed as described before [25,26]. In brief, cylinders of 0.6 mm were extracted from each patient’s primary, representative, formalin-fixed, and paraffin-embedded tumour block. The cylinders were assembled in an empty array recipient block [26]. As internal controls, both TMA blocks contained various control tissues, including normal colorectal tissue.

### 2.3. Immunohistochemistry

Freshly cut TMA blocks were further analysed. All slides were deparaffinized for immunohistochemical analysis and were subjected to heat-induced antigen retrieval in an autoclave at 121 °C for 10 min in citrate buffer at pH 9.0. 

Standard indirect immunoperoxidase procedures were used for CD10 (abcam, Cambridge, UK, clone SPM171, dilution 1:150), β-catenin (Agilent, Santa Clara, CA, USA, clone: β-catenin-1), Ki67 (abcam, Cambridge, UK, clone SPM171, dilution 1:150), p53 (Oncogene, Cambridge, MA, USA clone: DO1, murine monoclonal IgG_2a_, dilution 1:3600), MLH1 (Agilent, Santa Clara, CA, USA, clone ES05, 1:10 dilution), and MSH2 (Agilent, Santa Clara, CA, USA, clone FE11, dilution 1:200). Diaminobenzidine was used as a chromogen and sections were counterstained with Mayer’s hematoxylin.

CD10 immunostaining was typically cytoplasmic. The percentage of positive cells for tumour tissue was estimated and the staining intensity was recorded as 0, 1+, 2+, or 3+. For statistical analyses, the staining results were categorized into three groups, as previously described [27]. Tumours without any staining were considered negative. Tumours showing at least weak CD10 staining were considered positive. Tumours with 1+ or 2+ positivity in up to 50% of cells, or 3+ positivity in up to 20% of cells, were considered weakly positive. Tumours with 2+ staining in >50% or 3+ staining in >20% of cells were considered strongly positive. For β-catenin expression, membranous and nuclear staining intensity was categorized separately in 0 (negative), 1+ (weak), 2+ (moderate), or 3+ (strong).

For Ki67 evaluation, the number of invasive cancer cell nuclei that were positive for Ki67 immunostaining was divided by the total number of invasive cancer cell nuclei. Spots were stratified into Ki67 < 10% positive and Ki67 ≥ 10% positive. Nuclear p53 staining, MSH2, MLH1, and BRAF were stratified into positive and negative spots.

### 2.4. Statistics

Statistical analyses were performed with JMP^®^ 10.0.2 software (2012 SAS Institute Inc., Cary, NC, USA). Contingency tables and the χ^2^-test were performed to find associations between molecular parameters and tumour phenotype. Survival curves were calculated according to Kaplan–Meier. The Log-Rank test was applied to detect significant survival differences between groups. Cox’s proportional hazards regression analysis was performed to test the statistical independence and significance between pathological and clinical variables.

## 3. Results

### 3.1. Technical Implications

A total of 1711 tumour samples were analysed, of which 1469 (85.9%) were interpretable regarding CD10 expression. Reasons for non-informative cases (242, 14.1%) included lack of tissue samples or absence of unequivocal cancer tissue.

### 3.2. Clinicopathological Parameters

Gender was equally distributed, with 858 (50.1%) female and 853 (49.9%) male patients (Table 1). The mean age was 69 years (29–69). The majority of patients were diagnosed with a moderate differentiation (*n* = 1505, 88.0%) in tumour stage pT3 (*n* = 1104, 64.5%) and a tubular carcinoma (*n* = 1644, 96.0%), while about half of the cohort was nodal negative (*n* = 889, 52.0%). In total, 24.2% of tumours were localized in the right colon, 8.7% in the transverse colon, 30.0% in the descending colon, and 37.2% in the rectum.

### 3.3. CD10 Expression in Colorectal Cancer Cells 

CD10 expression was considered negative in 72.6%, weak in 20.1%, and strong in 7.4% of 1469 interpretable CRC patients (Table 2). A representative picture of CD10 immunostaining is depicted in Figure 1. CD10 expression was significantly associated with earlier tumour stages (*p* = 0.017). In total, 37.8% of pT1 tumours were positive for CD10, while the proportion decreased with increasing infiltration depth (pT2—33.4%, pT3—26.8%, pT4—21.7%). Interestingly the fraction of strongly expressing CD10 tumours was highest at the pT2 stage (11.3%). Tumour location was also significantly associated with the level of CD10 expression (*p* < 0.001). While CRC in the ascending colon or the transverse colon expressed CD10 in 18.6% and 15.9%, it was detectable in 34% of left-sided CRC and 28.1% of rectal cancer specimens. In contrast, CD10 expression was unrelated to the nodal status (*p* = 0.711) and tumour grading (*p* = 0.397). 

### 3.4. Association of CD10 Expression and Molecular Marker Expression

CD10 was significantly associated with higher nuclear β-catenin expression levels (*p* < 0.001, Table 3), degree of p53 positivity (*p* = 0.013,) and MLH positivity (*p* < 0.001), as well as BRAF negativity (*p* = 0.001). In contrast, membranous ß-catenin expression, Ki67 positivity > 10%, and MSH did not significantly correlate with CD10 expression levels. Representative pictures of Ki67, p53, and BRAF immunostaining are depicted in Figure 2.

### 3.5. Survival Analysis

Advanced tumour stages, high grading, and advanced nodal status were associated with poor patient overall survival (Figure 3a–c), as expected; while CD10 expression did not correlate with either overall survival (Figure 3d; *p* = 0.562) or survival in a subgroup analysis for tumour stage and nodal status (Figure 4a–f).

## 4. Discussion

CD10 expression in CRC cells is associated with earlier tumour stages and left-sided CRC, while grading and lymphatic spread do not correlate with CD10 expression levels. Furthermore, CD10 expression has no impact on patients’ overall survival, even when subgroup analyses for tumour stage or nodal status were evaluated. Therefore, our data exclude CD10 as a relevant prognosticator in CRC. 

Several studies, mainly conducted in Japan, have reported CD10 expression in CRC with conflicting results. Some data suggest an increase in tumoural CD10 expression in advanced CRC tumour stages, based on Duke’s classification [14,19], lymphatic invasion [20], lymph node metastasis [9,14,20], and liver metastasis [8,15,16,17,18], while others found no association with histopathological features such as tumour size or invasion depth [9,15,20,21,22], TNM staging [9], or lymph node or distant metastasis [9,15,21,22,28]. The majority of these studies are Asian single-centre cohorts, ranging in size from 30 to a few hundred patients. Our data, comprising 1469 CRC patients, reveal a significant association of CD10 expression with earlier tumour stages, with the highest proportion of CD10 positive tumours in the pT1 stage and the highest frequency of strong expression in the pT2 stage. Earlier studies discussed a functional role of CD10 in the early stages of carcinogenesis, promoting cell motility, tumour cell invasion, and dedifferentiation [8,14,16,22,23]. CD10 tissue expression, which is completely absent in healthy colorectal tissue [4,21], has been shown to consistently increase with the grade of dysplasia, ranging from low-grade adenoma to submucosal carcinomas [21,22,24].

In this context, CD10 has been discussed as promoter of a more aggressive tumour biology in early carcinogenesis, leading to more advanced tumour stages and distant metastasis [14,22]. However, in our large European cohort, we were able to exclude any association of CD10 expression with advanced tumour stages, such as lymph node metastasis, vascular invasion, or grading. Even in the pT1 stages, where CD10 has been shown to be an independent predictor of lymphatic spread [20], we found CD10 expression to be more frequent in nodal-negative patients. However, the authors noted a high risk of bias in their cohort due to a higher proportion of patients with lymph node metastases.

Furthermore, overall survival did not correlate with CD10 expression in this large European cohort, while grading, tumour, and nodal stage revealed a significant association with a poorer overall survival. Even in subgroup analyses for T-stage or lymph node involvement, CD10 could not discriminate populations with favourable survival. Thus, we can exclude CD10 as a relevant independent biomarker for aggressive CRC subtypes, which is in line with previous findings [9,22,23].

The grade of dedifferentiation did not show a significant association with CD10 expression, in our cohort, of invasive carcinomas. This, too, matches previous findings [15,21,28], while one cohort demonstrated an inverse correlation with tumour grading [9]. Without reaching statistical significance, our data revealed the highest proportion of CD10 positivity in G1 tumours, while G2 tumours showed the greatest ratio of strong CD10 expression. This supports the potential role of CD10 in early tumour progression, while not being relevant as a biomarker for aggressive tumour subtypes, at a later stage.

Further correlations with histopathological features in the CRC cohort suggest associations of CD10 expression with aggressive CRC phenotypes, i.e., p53 positivity, which indicates accumulation of p53 mutations, as mutant p53 is more stable and, therefore, accessible for immunohistochemical staining [29], and nuclear β-catenin expression. p53 mutations are among the most common genetic alterations found in human tumours and are detected in up to 43% of CRCs. They exert their tumorigenic effect both by reducing the tumour-suppressing activity of wild-type p53 and by conferring neomorphic functions, such as cell proliferation, cancer cell invasion, and cancer cell stemness [30,31]. Mutations in the Wnt/β-catenin pathway, which are also common in sporadic CRC, stabilize β-catenin, allowing its accumulation and translocation to the nucleus, where it acts as an activating transcription factor involved in cell proliferation and transmission [31]. Both, p53 and nuclear β-catenin are associated with tumour progression in sporadic CRC and are negative prognostic factors for disease-free and overall survival in CRC [29,32]. As previous studies and our data demonstrate, CD10 is expressed in early CRC carcinogenesis [21,22,24] and disappears in the more advanced tumour stages. In these early tumour stages, functional studies suggest that CD10 supports cancer progression by inhibiting apoptosis, enhancing cell motility, stimulating invasion, and promoting angiogenesis and dedifferentiation [8,14,16,22,23,33,34]. Although we can exclude CD10 as an independent prognosticator for CRC, CD10 expression is significantly associated with the indicators of aggressive CRC tumour biology, p53 mutations and nuclear β-catenin expression. In contrast, indicators for a deficient mismatch repair system (dMMR), such as MSH2, MLH1 negativity, and BRAF positivity, are not associated with CD10 expression. dMMR may favour tumour mutations, especially in the microsatellites, which have the potential to directly and indirectly affect coding DNA regions. Therefore, a high amount of neoantigens can be produced and make these highly microsatellite instable (MSI-H) tumours easier targets for the immune system, associated with a better prognosis. Germline mutations of MMR, most commonly inactivating mutations of MLH1 or MSH2, induce the most common tumour predisposition syndrome, known as the Lynch syndrome [35]. Both MLH1 and MSH2 negativity are not associated with CD10 expression. MSI-H can also occur in sporadic CRC, induced by MLH1 inactivity and somatic BRAF mutations [35]. However, BRAF-positivity is also not associated with CD10 expression. Overall, we find no evidence for an association of CD10 expression with dMMR and microsatellite instability, in our data.

The strength of this study is the large cohort size of 1469 consecutive CRC patients, while the retrospective retrieval of data and a loss to follow-up of 13.0% of the patients limit this study. In addition, the uneven distribution of T stages within our collective limits the significance; we were only able to include 53 patients (3.6%) in stage T1. However, our inclusion of T1 patients represents the third largest published collective of T1 patients in this context, apart from two other studies [20,22], and our main findings are in line with these publications. Moreover, p53 mutation was detected using immunohistochemistry and not confirmed using other methods.

## 5. Conclusions

Overall, our data indicate a correlation of CD10 expression with earlier tumour stages and left-sided colorectal cancer. Combined with the association of CD10 expression with indicators of aggressive tumour biology, our data support the idea of a potential functional role of CD10 in the early carcinogenesis of CRC. However, we can exclude CD10 as a relevant independent biomarker for CRC prognosis.

## Figures and Tables

**Figure 1 cancers-16-01473-f001:**
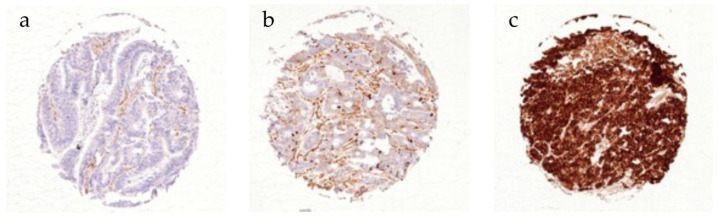
Immunohistochemical staining of CD10 in CRC. (**a**) Negative, (**b**) weak, and (**c**) strong CD10 expression.

**Figure 2 cancers-16-01473-f002:**
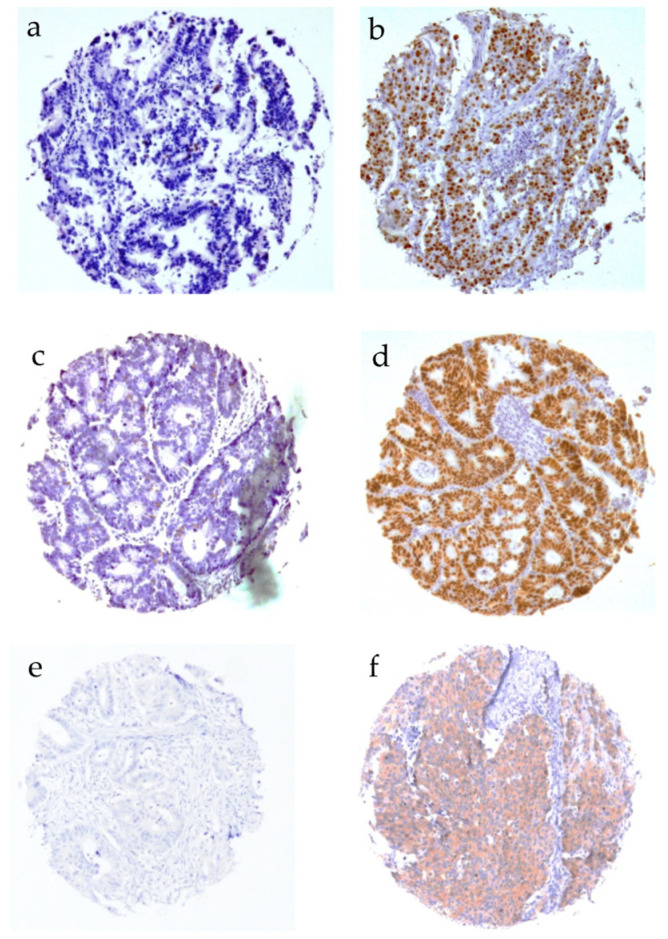
Immunohistochemical staining of Ki67, p53, and BRAF. (**a**) Ki67 < 10%, (**b**) Ki67 > 10%, (**c**) p53 positive, (**d**) p53 negative, (**e**) BRAF positive, and (**f**) BRAF negative.

**Figure 3 cancers-16-01473-f003:**
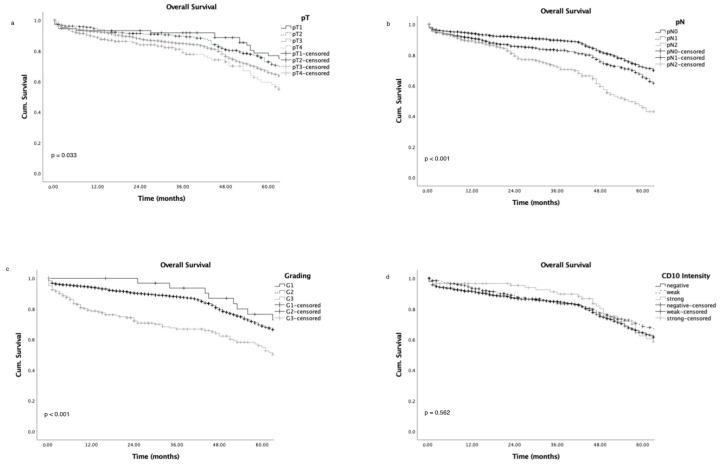
Overall survival curves according to tumour stage (**a**), nodal status (**b**), grading (**c**), and CD10 intensity level (**d**).

**Figure 4 cancers-16-01473-f004:**
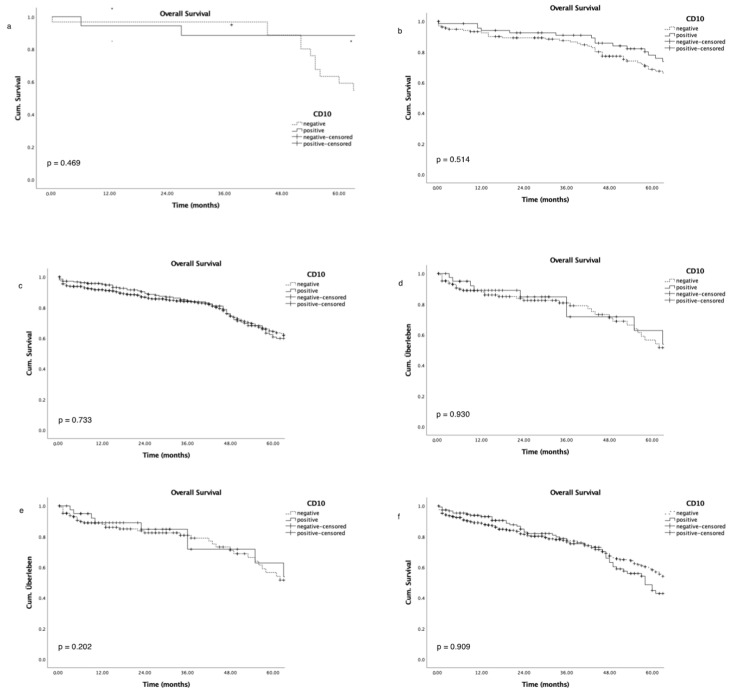
Association of overall survival and CD10 expression in subgroup analysis of tumour stage (pT1–pT4, (**a**–**d**)) and nodal stage (pN0–pN1–3, (**e**,**f**)).

**Table 1 cancers-16-01473-t001:** Clinical and pathological features of patients with colorectal cancers.

Parameter		*n* (%)
		*n* = 1711
Gender	female	858 (50.1)
	male	853 (49.9)
Age	(years)	68.8 (11.9)
Tumour grade	G1	29 (1.7)
	G2	1505 (88.0)
	G3	177 (10.3)
Tumour stage	pT1	75 (4.4)
	pT2	270 (15.8)
	pT3	1104 (64.5)
	pT4	262 (15.3)
Nodal status	pN0	889 (52.0)
	pN1	458 (26.8)
	pN2	364 (21.3)
Tumour type	tubular carcinoma	1644 (96.0)
	mucinous carcinoma	59 (3.4)
	others	8 (0.5)
Tumour localization	right colon	414 (24.2)
	transverse colon	149 (8.7)
	left colon	514 (30.0)
	rectum	637 (37.2)

**Table 2 cancers-16-01473-t002:** Association between CD10 expression level and standard histopathological parameter.

	CD10				
Parameter	*n*	Negative (%)	Weak (%)	Strong (%)	*p* Value
All Cancers	1469	72.6	20.1	7.4	
Tumour Stage					
pT1	53	62.3	32.1	5.7	**0.017**
pT2	221	66.5	22.1	11.3	
pT3	850	73.3	20.4	6.4	
pT4	198	78.5	16.1	5.6	
Lymph Node Metastasis					0.711
pN0	698	71.6	21.2	7.2	
pN1	329	75.7	17.4	7	
pN2	277	69.7	22.7	7.6	
pN3	1	100	0	0	
Grading					0.397
G1	21	81	14.3	4.8	
G2	1139	71.5	21.3	7.2	
G3	157	78.3	16	5.7	
Tumour Localization					**<0.001**
Right colon	258	81.4	12.8	5.8	
Transverse colon	88	84.1	11.4	4.5	
Left colon	250	66	26.4	7.6	
Rectum	416	71.9	22.6	5.5	
Histological Subtype					**0.048**
Adenocarcinoma	938	72.8	20.8	6.4	
Mucinous	67	91	6	3	
Others	9	66.7	33.3	0	
Peritumoural Lymphocytes					0.871
Absent	567	74.3	19.4	6.3	
Present	438	74	20.3	5.7	
Vascular Invasion					0.376
No	565	74.7	18.6	6.7	
Yes	439	73.3	21.5	5.2	

Data are displayed as percentages, if not otherwise indicated. *p*-values in bold indicate statical significance. n—number.

**Table 3 cancers-16-01473-t003:** Association between CD10 expression level and molecular marker.

	CD10				
Parameter	n Evaluable	Negative (%)	Weak (%)	Strong (%)	*p* Value
ß-catenin membranous					0.070
negative	110	80.0	12.7	7.3	
weak	328	77.7	12.8	9.5	
moderate	93	65.6	17.2	17.2	
strong	329	69.9	17.0	13.1	
ß-catenin nuclear					<**0.001**
negative	444	80.4	11.5	8.1	
weak	157	68.8	20.4	10.8	
moderate	140	70.7	13.6	15.7	
Strong	130	58.5	21.5	20.0	
Ki67					0.085
<10%	501	75.8	13.8	10.4	
≥10%	776	70.2	17.4	12.4	
p53					**0.013**
negative	374	77.5	13.9	8.6	
positive	270	68.5	15.9	15.6	
MSH2					0.212
negative	190	78.4	12.1	9.5	
positive	454	71.8	15.9	12.3	
MLH1					**<0.001**
negative	128	87.5	5.5	7.0	
positive	515	70.3	17.1	12.6	
BRAF					**0.001**
negative	1141	69.1	17.2	13.7	
positive	100	84.0	13.0	3.0	

Data are presented as percentages. *p*-values in bold indicate statical significance between CD10 expression levels.

## Data Availability

The data presented in this study are available on request from the corresponding author.

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
