# Peer review of "CD10 Expression Correlates with Earlier Tumour Stages and Left-Sided Tumour Location in Colorectal Cancer but Has No Prognostic Impact in a European Cohort"

_cancers, 2024, doi:10.3390/cancers16081473_

Round 1

Reviewer 1 Report

Comments and Suggestions for Authors

The strength of thid study is the big amount of studied patients. The study is well designed and results are well planned and interesting. According with the study, CD10 expression correlates with earlier tumour stages, colon cancer location and indicators of aggressive CRC subtypes. However, they can exclude CD10 as a relevant independent prognosticator for CRC.

Author Response

We thank Reviewer 1 for the appreciation of our work.

Reviewer 2 Report

Comments and Suggestions for Authors

In the present work, the authors assessed the association of CD10 expression with tumor phenotype, molecular features, prognosis, p53 and nuclear b-catenin expression, and, the overall survival, in CRC. The authors concluded that the expression of CD10 in CRC cells is associated with earlier tumor stages and left-sided CRC, whereas lymph node involvement, tumor grading, and OS did not correlate with CD10 expression levels. 

This manuscript has several major and minor drawbacks that must be improved and corrected.

1) The biggest drawback of the present study is the small number of CRC tumor samples with earlier stages, accounting for only 3.6% (53) of the total number. Therefore, under these conditions, this means nothing. In other words, this finding does not support the idea of a functional role of CD10 in the early stages of carcinogenesis.

2) The minor drawback of the present study is that the authors pointed out the correlation of the expression of CD10 with the earlier tumor stages and reported that of the total tumor analyzed samples, 85.9% (1469) were interpretable regarding CD10 expression, due to lack of tissue samples or absence of unequivocal cancer tissue. This is different for lane n. 186.

Author Response

In the present work, the authors assessed the association of CD10 expression with tumor phenotype, molecular features, prognosis, p53 and nuclear b-catenin expression, and, the overall survival, in CRC. The authors concluded that the expression of CD10 in CRC cells is associated with earlier tumor stages and left-sided CRC, whereas lymph node involvement, tumor grading, and OS did not correlate with CD10 expression levels.

This manuscript has several major and minor drawbacks that must be improved and corrected.

1) The biggest drawback of the present study is the small number of CRC tumor samples with earlier stages, accounting for only 3.6% (53) of the total number. Therefore, under these conditions, this means nothing. In other words, this finding does not support the idea of a functional role of CD10 in the early stages of carcinogenesis.

We thank Reviewer 2 for the chance to improve our work.

We discussed this aspect more critically as follows:

“Earlier studies discussed a functional role of CD10 in early stages of carcinogenesis promoting cell motility, tumor cell invasion and dedifferentiation.”

Additionally, we discussed the uneven distribution of T-stages in the Limitations:

“In addition, the uneven distribution of T stages within our collective limits the significance: we were only able to include 53 patients (3.6%) in stage T1. However, our inclusion of T1 patients represents the third largest published collective of T1 patients in this context, apart from two other studies [16, 21], and our main findings are in line with these publications.”

2) The minor drawback of the present study is that the authors pointed out the correlation of the expression of CD10 with the earlier tumor stages and reported that of the total tumor analyzed samples, 85.9% (1469) were interpretable regarding CD10 expression, due to lack of tissue samples or absence of unequivocal cancer tissue. This is different for lane n. 186.

We changed the patient number in line 186 from 1711 to 1469.

Reviewer 3 Report

Comments and Suggestions for Authors

Authors of the submited manuscript investigated the expression of CD10 marker in colorectal cancer tissue and have found that CD10 expression is associated with earlier tumor stages, left-sided tumors but can be excluded as a relevant independent prognostic factor for colorectal cancer. Authors performed analysis on 1711 cancer patients aiming to clarify controversis on potential application of CD10 as a tumor marker in this disease.

The paper is clearly written and data presented, and there are only few suggestions for the minor changes.

Although initially 1711 were involved, results were obtained and analysed for 1469 samples, as authors stated that „reasons for non-informative cases (242, 14.1 %) included lack of tissue samples or absence of unequivocal cancer tissue“ – thus, the actual number of samples was 1469 and only that number should be given in the manuscript.

There is a certain degree of repetition of literature data in Introduction and Discussion – the information should be removed from either part. Also, some other relevant articles were omitted and should be included and discussed:

1.Del Rio et al. Is CD10 a reliable marker of invasive colorectal cancer? Ann. Ital. Chir. 2011 82: 279-282.

2.De Oliveira et al. Tissue expression of CD10 protein in colorectal carcinoma: correlation with the anatomopathological features of the tumor and with lymph node and liver metastases. J. Coloproctol. 2012, 32: 34-39.

3.Magadhi et al. Stromal Expression of CD10 in Colorectal Carcinoma and its Correlation with Lymph Node Metastasis and Other Prognostic Factors. J. Clin. Diagn. Res. 2019, 13: EC01-EC05.

Technical errors:

-page 3, line 103: (pH 9.0)

-page 3, line 134: Table 1. Clinical and pathological features of PATIENTS WITH colorectal cancer

-page 4, Table 2, second column: n (remove „evaluable“)

Comments on the Quality of English Language

-

Author Response

Authors of the submitted manuscript investigated the expression of CD10 marker in colorectal cancer tissue and have found that CD10 expression is associated with earlier tumor stages, left-sided tumors but can be excluded as a relevant independent prognostic factor for colorectal cancer. Authors performed analysis on 1711 cancer patients aiming to clarify controversies on potential application of CD10 as a tumor marker in this disease.

The paper is clearly written and data presented, and there are only few suggestions for the minor changes.

Although initially 1711 were involved, results were obtained and analysed for 1469 samples, as authors stated that „reasons for non-informative cases (242, 14.1 %) included lack of tissue samples or absence of unequivocal cancer tissue“ – thus, the actual number of samples was 1469 and only that number should be given in the manuscript.

            We thank Reviewer 3 for this appreciative review. We corrected the patient number within the entire manuscript.  

There is a certain degree of repetition of literature data in Introduction and Discussion – the information should be removed from either part. Also, some other relevant articles were omitted and should be included and discussed:

1.Del Rio et al. Is CD10 a reliable marker of invasive colorectal cancer? Ann. Ital. Chir. 2011 82: 279-282.

2.De Oliveira et al. Tissue expression of CD10 protein in colorectal carcinoma: correlation with the anatomopathological features of the tumor and with lymph node and liver metastases. J. Coloproctol. 2012, 32: 34-39.

3.Magadhi et al. Stromal Expression of CD10 in Colorectal Carcinoma and its Correlation with Lymph Node Metastasis and Other Prognostic Factors. J. Clin. Diagn. Res. 2019, 13: EC01-EC05.

            We removed the literature details from the introduction and included the article of Del Rio (1.) in the manuscript. The findings of these studies were consistent with the aforementioned literature, so no adaption of the manuscript was necessary.

The other two articles are published in non-pubmed journals and do not provide new information on tumour CD10 expression. The last article by Magadhi et al. (3.) deals with stromal expression of CD10, which was not analysed in our study. Therefore, we did not considere these two publications as references.

Technical errors:

-page 3, line 103: (pH 9.0)

-page 3, line 134: Table 1. Clinical and pathological features of PATIENTS WITH colorectal cancer

-page 4, Table 2, second column: n (remove „evaluable“)

            We corrected these errors.

Round 2

Reviewer 2 Report

Comments and Suggestions for Authors

1) The authors suggested an association between CD10 expression and aggressive CRC phenotypes based on p53 mutant positivity and nuclear β-catenin; However, the data reported here, based on the expression of CD10, do not correlate with lymph node metastasis, vascular invasion, or grading (features which are describing the aggressive form of CRC), but instead with early tumor stage. In addition, I wondered whether the authors had confirmed the mutation in the p53 gene.

2) The authors reported a positive correlation between the expression levels of CD10 and nuclear ß-catenin, p53 and MLH positivity, and BRAF negativity. Please do revise discussion once more.

3) Could you also add some representative immunohistochemical images of p53, Ki67, MSH, MLH, and BRAF.

Author Response

1) The authors suggested an association between CD10 expression and aggressive CRC phenotypes based on p53 mutant positivity and nuclear β-catenin; However, the data reported here, based on the expression of CD10, do not correlate with lymph node metastasis, vascular invasion, or grading (features which are describing the aggressive form of CRC), but instead with early tumor stage. In addition, I wondered whether the authors had confirmed the mutation in the p53 gene.

            We thank Reviewer 2 for this comment and clarified this point in the discussion as follows:

As previous studies and our data demonstrate, CD10 is expressed in early CRC carcinogenesis [21, 22, 24] and disappears in more advanced tumor stages. In these early tumor stages, functional studies suggest that CD10 supports cancer progression by in-hibiting apoptosis, enhancing cell motility, stimulating invasion and promoting angio-genesis and dedifferentiation [8, 14, 16, 22, 23, 33, 34]. Although we can exclude CD10 as an independent prognosticator for CRC, CD10 expression is significantly associated with the indicators of aggressive CRC tumor biology p53 mutations and nuclear β-catenin expression.”

p53 was detected by immunohistochemistry and not confirmed by other methods as described. To clarify this, we added the following sentence to the limitation section:

“Moreover, p53 mutation was detected by immunohistochemistry and not confirmed by other methods.”

2) The authors reported a positive correlation between the expression levels of CD10 and nuclear ß-catenin, p53 and MLH positivity, and BRAF negativity. Please do revise discussion once more.

We revised the discussion as follows:

“p53 mutations are among the most common genetic alterations found in human tumors and are detected in up to 43% of CRCs. They exert their tumorigenic effect both by re-ducing the tumour suppressing activity of wild-type p53 and by conferring neomorphic functions, such as cell proliferation, cancer cell invasion and cancer cell stemness [30, 31]. Mutations in the Wnt/β-catenin pathway, which are also common in sporadic CRC, sta-bilize β-catenin, allowing its accumulation and translocation to the nucleus, where it acts as an activating transcription factor involved in cell proliferation and transmission[31]. Both, p53 and nuclear β -catenin are associated with tumour progression in sporadic CRC and are negative prognostic factors for disease-free and overall survival in CRC [29, 32]..”

and

“In contrast, indicators for a deficient mismatch repair system (dMMR), such as MSH2, MLH1 negativity and BRAF positivity are not associated with CD10 expression. dMMR may favor tumor mutations especially in the microsatellites, which have the potential to directly and indirectly affecting coding DNA regions. Thereby, a high amount of neo-antigens can be produced and make these highly microsatellite instable (MSI-H) tumors easier targets for the immune system associated with a better prognosis. Germline mu-tations of MMR, most commonly inactivating mutations of MLH1 or MSH2, induce the most common tumor predisposition syndrome known as the Lynch syndrome [35]. Both, MLH1 and MSH2 negativity are not associated with CD10 expression. MSI-H can also occur in sporadic CRC induced by MLH1 inactivity and a somatic BRAF-mutations [35]. Though, BRAF-positivity is also not associated with CD10 expression. Overall, we find no evidence for an association of CD10 expression with dMMR and microsatellite instability in our data.”

3) Could you also add some representative immunohistochemical images of p53, Ki67, MSH, MLH, and BRAF.

We added a picture of p53, Ki67 and BRAF to the manuscript as Figure 2.

Round 3

Reviewer 2 Report

Comments and Suggestions for Authors

All comments have been provided according to the reviewers’ suggestions; therefore, the revised manuscript can be accepted in the present form.